# Amplification of Hippo Signaling Pathway Genes Is Governed and Implicated in the Serous Subtype-Specific Ovarian Carcino-Genesis

**DOI:** 10.3390/cancers16091781

**Published:** 2024-05-05

**Authors:** Karthik Balakrishnan, Yuanhong Chen, Jixin Dong

**Affiliations:** Eppley Institute for Research in Cancer and Allied Diseases, Fred & Pamela Buffett Cancer Center, University of Nebraska Medical Center, Omaha, NE 68198, USA; kbalakrishnan@unmc.edu (K.B.); cheny@unmc.edu (Y.C.)

**Keywords:** ovarian cancer, serous subtype, Hippo signaling pathway, amplification

## Abstract

**Simple Summary:**

This work investigated the impact of Hippo signaling pathways in the development of ovarian cancer. In order to investigate, the signatures relevant to Hippo signaling pathway were collected from the molecular signatures database, and the expression patterns of these signatures were explored in the mRNA expression profiles of ovarian cancer cohorts. The findings of this study demonstrate that the Hippo signaling pathway signatures are dysregulated significantly in serous subtype-specific ovarian carcinogenesis compared to other subtype ovarian cancers. The receiver operating characteristic curve using the Hippo gene set and its associated gene expressions could predict the serous subtype cancers with greater specificity and sensitivity. Furthermore, the cBioPortal database’s high-grade serous ovarian cancer profile examined the Hippo gene set for mutation analysis. The Hippo signaling pathway genes are amplified significantly in grade three and stage three or severe fourth-type ovarian cancers. Furthermore, the Dependency Map plot results prove that these genes are amplified extensively across ovarian cancer cell lines. Additionally, analyses of the overall survival curve plot results also demonstrated that these gene expressions exhibit poor survival patterns associated with highly expressed conditions of serous subtypes of ovarian cancer patients.

**Abstract:**

Among women, ovarian cancer ranks as the fifth most common cause of cancer-related deaths. This study examined the impact of Hippo signaling pathway on ovarian carcinogenesis. Therefore, the signatures related to Hippo signaling pathway were derived from the molecular signatures database (MSigDB) and were used for further analysis. The Z score-based pathway activation scoring method was employed to investigate the expression patterns of these signatures in the mRNA expression profiles of ovarian cancer cohorts. Compared to other subtype tumors, the results of this study show that the Hippo signaling pathway signatures are dysregulated prominently in serous subtype-specific ovarian carcinogenesis. A receiver operating characteristic (ROC) curve-based results of the Hippo gene set, yes-associated protein 1 (YAP1), and mammalian sterile 20-like kinases 1 (MST1) genes can predict the serous subtype tumors by higher specificity and sensitivity with significant areas under the curve values also further reconfirmed these signaling dysregulations. Moreover, these gene sets were studied further for mutation analysis in the profile of high-grade serous ovarian adenocarcinoma in the cBioPortal database. The OncoPrint results reveal that these Hippo signaling pathway genes are amplified highly during the grade three and stage third or fourth of serous type ovarian tumors. In addition, the results of the Dependency Map (DepMap) plot also clearly show that these genes are amplified significantly across the ovarian cancer cell lines. Finally, overall survival (OS) curve plot investigations also revealed that these gene expressions show poor survival patterns linked to highly expressed conditions in serous subtypes of ovarian cancer patients with significant *p*-values (*p* < 0.05). Thus, the current finding would help to develop the targeted therapies treatment for serous subtype ovarian carcinogenesis.

## 1. Introduction

The ovary is a dynamic organ that changes dramatically during each reproductive cycle. For effective reproduction, the ovary must undergo this cyclical remodeling, which includes the processes of follicle genesis, ovulation, and development and regression of the luteal phase [1,2]. Ovarian cancer ranks fifth among women’s cancer-related fatalities and is also the deadliest gynecological disease in the US [3]. Every year globally, about 225,500 women are diagnosed with ovarian cancer, and among these, 140,200 fatalities occur [4]. Ovarian malignancies are divided into high-grade and low-grade serous carcinomas [5]. A small portion of tumors are low-grade serous carcinomas, while 80% of ovarian epithelial carcinomas are high-grade serous tumors. The high-grade tumors differ from other subtypes of ovarian cancer due to their aggressiveness, higher mutation rate, and oncogene amplification [6]. High-grade serous tumors are poorly understood in terms of their place and origin of cells. It can be challenging to determine the disease’s origin since, in most cases, the cancer cells have already extensively spread to other tissues or organs when the cancer is discovered [7]. The significant challenge of these tumors is the lack of biomarkers for the early detection of precursor lesions. Later, it was identified that ovarian surface epithelial cells may have undergone a neoplastic transformation to become high-grade serous tumors [8]. Patients with high-grade serous cancers typically have lower survival rates [4].

Hippo signaling has emerged as a critical pathway in cancer cells. This signaling has downstream transcriptional coactivators, such as yes-associated protein 1 (YAP1) and transcriptional coactivator with PDZ-binding motif (TAZ), two homologs of Yorkie (Yki) found in Drosophila [9]. Tissue homeostasis and development depend on canonical pathways via these signaling core components (YAP1 and TAZ), whereas aberrant signaling has been accompanied by several diseases, including cancer [10]. Many cancers, including lung, breast, melanoma, prostate, and colorectal cancer, have widespread YAP1/TAZ activation [11], and this activation is crucial for the initiation, development, and metastasis of cancer. The uncontrolled activation of YAP1/TAZ has been linked to developing malignant features, such as cancer stem cell self-renewal, metastasis, and drug resistance [12,13]. Some studies reported that YAP1/TAZ activation causes drug resistance by downregulating various multidrug transporters [14,15].

The current study used integrative functional genomic approaches to investigate the Hippo signaling pathway dysregulations across the various subtypes comprising ovarian cancer mRNA expression profiles and their cell line profiles. Most of the Hippo signaling and their downstream target pathway signatures have been highly dysregulated in serous subtype-specific ovarian tumors compared to other subtypes of ovarian cancer. Moreover, exploring mutation analysis, ROC curves, and overall survival curve plot results further reconfirm this subtype-specific ovarian carcinogenesis. Thus, serous subtype-specific Hippo signaling pathways dysregulation has been identified, and this finding information might help to develop the targeted therapeutics treatment of this subtype-specific ovarian tumors.

## 2. Materials and Methods

### 2.1. Genome-Wide Expression Profiles 

The gene expression omnibus (GEO), a genome-wide expression archive, offered mRNA expression profiles of ovarian cancer for this study [16]. In the present work, the many subtypes consisting of mRNA expression profiles of the ovarian cancer cohorts GSE6008, GSE10971, GSE14407, and GSE12470 were employed. Further validation was also performed using the ovarian cancer cell lines profile GSE29175. The data are either in raw .CEL format or are normalized files obtained from GEO, while the Affy package used for RMA/MAS 5.0 normalizes these raw files [17]. Thus, mRNA expression profiles were gathered and employed for further studies. 

### 2.2. Pathway Activation Scoring 

Most of the traits in our body are complicated and not controlled by a single gene. A group of genes or gene sets governs most biological pathways, processes, and activities. Thus, these gene sets were known as signatures [18]. The molecular signatures database (MSigDB) was used to gather the gene set exposed to various biological or experimental conditions [19]. The current study used ovarian cancer mRNA expression profiles to score and analyze the activation patterns of the Hippo signaling pathways signatures gathered from MSigDB. In total, 13 signatures were gathered and utilized in this study. Many studies have already established and employed Z-score-based pathway activation prediction used in the study [20,21,22,23]. The STRING database (https://string-db.org/, accessed on 8 March 2024) also achieved protein–protein interaction [24].

### 2.3. Hierarchical Clustering Analysis

Hierarchical clustering is the primary statistical technique for determining groups of genes or samples with similar functions expressed and clustered in various biological situations [25]. The DNA-Chip (dChip) analyzer was the primary tool for hierarchical clustering analysis [26,27]. Higher gene expressions are displayed in red color, whereas lower expressions are in green color in the heatmap. The sample-specific clustered enrichment *p* values were determined using the hypergeometric distribution in the dChip tool. Moreover, R packages were used to perform the box plots [28,29]. The Hippo signaling downstream target genes, including yes-associated protein 1 (YAP1) and transcriptional coactivator with PDZ-binding motif (TAZ) expressions, were performed in ovarian cancer subtypes containing profiles. The significant *p* values are calculated by one-way ANOVA.

### 2.4. Receiver Operating Characteristic (ROC) Curve

Numerous tumor subtypes accompany ovarian cancers. The analysis of the Hippo signaling pathway genes, including mammalian sterile 20-like kinases 1, yes-associated protein 1, and Hippo gene set expression values or mean expression values given as input values in the following profiles: GSE10971 and GSE14407. A receiver operating characteristic (ROC) curves were plotted in the MedCalc software tool [30,31].

### 2.5. Mutation Analysis

cBioPortal (https://www.cbioportal.org/, accessed on 1 April 2024), a publicly accessible database for tumor genomics and transcriptomics, was used for performing an integrated functional genomics analysis in the RNA sequencing profile of high-grade serous ovarian adenocarcinoma [32]. Large-scale transcriptome and genomic data analysis and visual representation of tumor alterations were also feasible with this database. The Hippo gene set was used to explore the mutation levels in this database. Additionally, these gene sets were studied further via copy-number variation analysis across the ovarian cancer cell lines of the Dependency Map (DepMap) Portal (https://depmap.org/portal/, accessed on 5 March 2024) [33].

### 2.6. Survival Curve

Kaplan–Meier (KM) plots (http://kmplot.com) are used to investigate the survival curve analysis [34,35,36]. The Kaplan–Meier plot resources, which included follow-up clinical data and mortality rate information, were called upon to investigate the impact of specific gene expression on the survival pattern of patients with serous subtype ovarian cancer. Therefore, human Hippo genes mammalian sterile 20-like kinases 1/2 (MST1/2), yes-associated protein 1 (YAP1), and Hippo gene set were analyzed among the patients with serous subtypes. The log-rank test determined the *p*-values [37]. 

## 3. Results

### 3.1. Hippo Signaling Pathway Is Dysregulated Extensively in Serous Subtypes of Ovarian Cancer 

The current study explored the effects of Hippo signaling pathway dysregulations on ovarian carcinogenesis. Therefore, the 13 signatures representing the Hippo signaling pathway were derived from the molecular signatures database (Appendix A). Moreover, their expression patterns were studied in the mRNA expression profiles of various subtypes containing ovarian cancer, including non-cancerous, mucinous, endometrioid, clear cell, and serous samples profiles GSE6008, GSE10971, and GSE14407. The Z-score activation pattern revealed that Hippo signaling pathway is dysregulated significantly in serous-specific subtypes rather than other ovarian cancer subtypes with significant *p*-values (*p* < 0.05) (Figure 1A–C). Furthermore, the impact of these pathway dysregulations was further validated with ovarian cancer cell-line profiles GSE29175. The result also indicates that these signatures are prominently elevated across ovarian cancer cell lines (Appendix A). Additionally, the Hippo signaling pathway downstream genes, including YAP1 and TAZ expression patterns, were investigated in GSE6008 and GSE12470 for further confirmation. The boxplot results exhibited that these genes are intensely activated in serous or advanced serous subtype ovarian tumors with significant *p* values (*p* < 0.05) compared to other subtypes of ovarian tumors (Appendix A). Thus, Hippo signaling pathway dysregulation has been identified and plays a pivotal role in serous subtype-specific ovarian tumorigenesis.

### 3.2. Hippo Signaling Gene Set Is Also Greatly Enriched in Serous Subtypes

These signaling pathway dysregulations are validated further by identifying the genes as the Hippo gene set, of which the genes are consistently found among Hippo signaling pathway signatures (Appendix A**).** The derived Hippo gene sets were studied in the non-cancerous ovarian tissues and ovarian cancer subtypes encompassing mRNA expression profiles GSE6008, GSE10971, and GSE12470. The Hippo gene set is mainly expressed in serous subtype ovarian tumors (Figure 2A–C). Moreover, these gene sets were evaluated for their biological function in the gene ontological analysis. The result of the ontological functional exploration showed that it regulates the organs and growth development (Figure 2D). For additional confirmation, the Hippo gene sets were also studied using gene-set enrichment analysis (GSEA) in those profiles. The results of GSEA show significant enrichment scores for serous-type ovarian tumors (Figure 2E,F). Consequently, all these results further prove that this signaling pathway dysregulation may play critical roles in serous subtype-specific ovarian carcinogenesis.

### 3.3. Hippo Pathway Gene Expressions Are Showing Poor Prognosis with Greater Specificity and Sensitivity in Serous Subtype Ovarian Tumors

Another approach to confirm this pathway dysregulation by receiver operating curve (ROC)-based evaluation was accomplished in the following mRNA expression profiles: GSE10971 and GSE14407. Hence, the human Hippo gene mammalian sterile 20-like kinases 1 (MST1) and its downstream target gene yes-associated protein 1 (YAP1) expression were also validated along with the Hippo gene set in those profiles. The ROC curve results demonstrate that the Hippo gene set, YAP1, and MST1 genes could predict the serous subtype ovarian tumors with better specificity and sensitivity with significant areas under the curve (AUC) and *p* values (*p* < 0.05) (Figure 3A,B). These findings also reconfirmed that Hippo signaling pathways dysregulations might contribute to the carcinogenesis of serous-type ovarian tumors.

### 3.4. Hippo Gene Set Is Amplified Significantly in High-Grade Serous Ovarian Cancer 

Additionally, the impact of the Hippo gene set on high-grade serous ovarian adenocarcinoma profile TCGA through mutation analysis has been investigated. The OncoPrint results showed that all these genes had significantly amplified in high-grade serous carcinoma patients except the LATS1 gene (Figure 4A,B). These genes have also been amplified, particularly in grade three and stage three or four ovarian cancers of the serous type. An alternative method to confirm the amplification of these genes in ovarian cancer cell lines with copy-number variation analysis was also examined using the Dependency Map (depMap) Portal. As a result, these genes are amplified extensively throughout the ovarian cancer cell lines (Figure 5A–F). These results show that Hippo signaling-involved pathway genes are greatly amplified and are implicated during high-grade serous specific ovarian carcinogenesis.

### 3.5. Hippo Pathway Gene Expressions Are Also Associated with Poor Survival in Serous Ovarian Cancer Patients

Furthermore, the Hippo gene set was also explored for protein-protein interactions using the STRING database. The result shows that these proteins have been intensely correlating with higher co-expression patterns (Figure 6A). Moreover, the effect of these gene expressions on the lifetime of serous subtype ovarian cancer patients was also studied. Hence, Kaplan-Meier (KM) plots were used to examine these patients’ overall survival (OS) probability. The overall survival curve plots of the YAP1 gene and Hippo gene set expressions exhibit a poor survival pattern associated with highly expressed conditions along with significant *p*-values (*p* < 0.05) (Figure 6B–E). However, the MST1 and MST2 gene expressions did not impact the survival of serous subtypes of patients. Thus, the overall survival plot results also add further evidence that Hippo signaling pathways’ dysregulation also affects serous ovarian cancer patients’ OS rate.

## 4. Discussion

The set of kinases that made up the Hippo signaling pathway was identified in Drosophila, although it is highly conserved in many animals. Moreover, mammalian cells have also shown this cascade of kinases [38]. In the kinase cascade, YAP1 and TAZ are the two essential co-activators of transcription [39]. When Hippo signaling turns on, mammalian sterile 20-like kinases 1/2 (MST1/2) and its adaptors salvador homolog1 (SAV1) complex triggered phosphorylate the large tumor suppressor 1/2 (LATS1/2) and MOB kinase activators 1A and 1B (MOB1A/B) complex, subsequently phosphorylating YAP1/TAZ complex and then it degrades. On the other hand, YAP1/TAZ is activated in response to Hippo pathway suppression. Moreover, it then translocates to the nucleus, binds to the transcription-enhancing association domain (TEAD), and promotes TEAD-mediated gene transcription [40,41].

The intracellular or extracellular signals, such as G protein-coupled receptors (GPCRs), stress, cell polarity, and inter-cellular contact, could trigger the Hippo signaling pathway. Among these signals, GPCRs can activate or inactivate the Hippo/YAP1 signaling pathways [42]. Under normal conditions, the Hippo pathway might control proliferation, growth, and organ development, while abnormality in this pathway may lead to cancer [43]. YAP1 is identified as a cancer oncogene [44] and suggests that YAP1 contributes excessively to the progression of many cancers [45]. The persistent YAP1 activation increases ovarian cancer cell proliferation, migration, and resistance to cisplatin-mediated cellular apoptosis [46]. Furthermore, this higher expression was linked with poor survival in ovarian cancer patients. The overexpression of TAZ in ovarian cancer also stimulates cell migration, proliferation, and the epithelial–mesenchymal transition [47]. YAP1 was expressed highly in ovarian cancer patients with clear cell subtypes of ovarian tumors [44]. In the current study, our results reveal that the amplification of YAP1/TAZ and its expression are associated with high-grade serous subtype-specific ovarian tumor carcinogenesis.

One-third of several human cancers are accompanied by mutations in KRAS genes and their dysregulated signaling pathways [48]. In ovarian cancer, KRAS mutation was an increasing trend from normal ovaries to benign mucinous tumors, mucinous borderline tumors, and mucinous ovarian tumors [49,50]. KRAS mutations were more prevalent in particularly borderline ovarian tumors than malignant tumors [51]. Furthermore, KRAS mutations are more common in low-grade serous ovarian cancer along with BRAF mutations [52]. In contrast, NRAS mutation is the oncogenic driver for serous subtype ovarian carcinoma along with somatic mutation of TP53, germline mutations of BRCA1 and BRCA2, and lower frequencies of NF1, PTEN, and RB1 mutations [53,54]. Moreover, a previous study reports that YAP1’s transcriptional activity is modulated by oncogenic KRAS signaling through the MAPK pathway in pancreatic ductal adenocarcinoma without changing its subcellular location. It implies that KRAS can phosphorylate or use other posttranscriptional modifications mediated by MAPK and its downstream target of YAP1 without supporting the Hippo pathway [55]. YAP1 stimulated the production of genes encoding secretory factors that supported neoplastic proliferation and cancer progression.

The present study results also reveal that Hippo signaling pathway is highly dysregulated in serous-type ovarian tumors compared to other subtypes of ovarian tumors. Furthermore, these pathway genes are also amplified vastly and are implicated in the carcinogenesis of high-grade serous ovarian tumors. Moreover, these signaling pathway gene expressions also affect the overall survival rate of the corresponding subtype tumor patients. Thus, the current study identified serous subtype-specific signaling pathway dysregulations, and this lead information would help identify possible drugs for treating these malignancies.

## 5. Conclusions

The current study identified that Hippo signaling pathway is highly dysregulated in serous subtype-specific ovarian cancer rather than other subtypes of ovarian tumors. This pathway dysregulation in the subtypes of ovarian cancer has been identified from (i) the Hippo gene set exclusively expressed in the serous type rather than other subtypes of ovarian tumors, (ii) the ROC curve of these signaling pathway genes expressions also display better sensitivity and specificity in the corresponding subtypes, iii) these pathway genes also amplified vastly during the high grade serous ovarian tumor samples along with ovarian cancer cell lines, and iv) the genes expression of this pathway also associated with poor survival in serous subtype of ovarian tumor patients. However, further studies are required to draw conclusions.

## Figures and Tables

**Figure 1 cancers-16-01781-f001:**
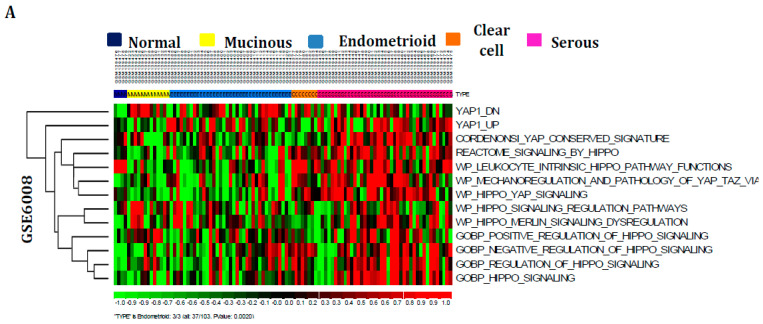
The impacts of Hippo signaling pathway dysregulations during ovarian oncogenesis were examined. (**A**–**C**) The molecular signatures database (MSigDB) was used to collect Hippo signaling pathway signatures, and the mRNA expression profiles of ovarian cancer GSE6008 (**A**), GSE10971 (**B**), and GSE14407 (**C**) were studied to determine their activation patterns. Compared to other ovarian cancer subtypes, the Z score activation pattern shows that Hippo signaling pathways are significantly more abundant in serous subtypes. The *p*-values for each subtype-specific enrichment of the samples were determined using the hypergeometric distribution.

**Figure 2 cancers-16-01781-f002:**
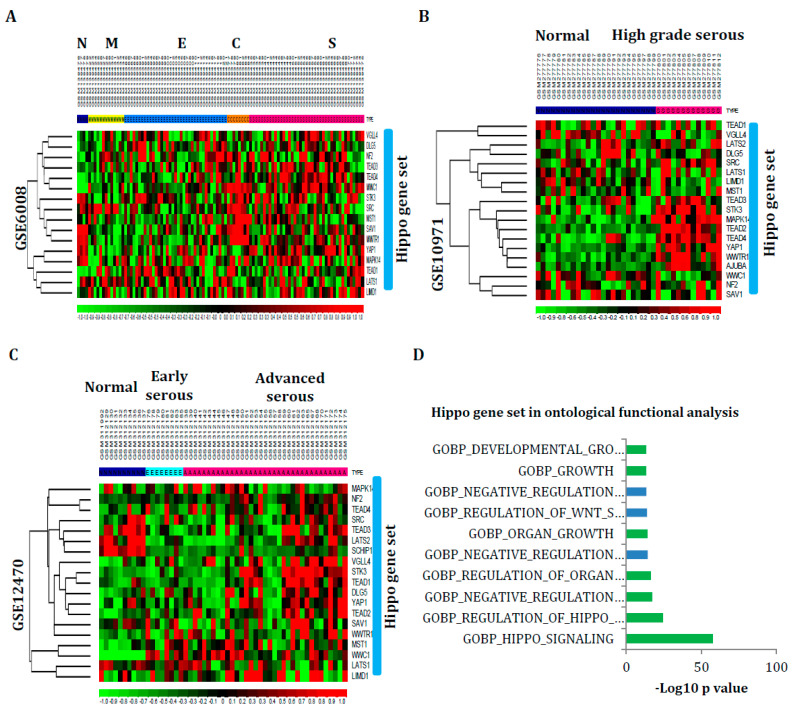
Hippo signaling genes or sets of genes frequently found across the Hippo pathway signatures were investigated in non-cancerous ovarian tissues and ovarian cancer subtypes comprising profiles GSE6008, GSE10971, and GSE12470. (**A**–**C**) The Hippo gene set is expressed more predominantly in serous-type ovarian cancers than in other subtype tumors. (**D**) The result of the ontological functional analysis revealed that this gene set governs the development of organs and growth. Furthermore, this gene set was also examined using gene-set enrichment analysis (GSEA) for additional confirmation (Green color indicates the ontology function of the hippo gene set). (**E**,**F**) The GSEA results show substantial enrichment scores for serous subtype ovarian tumors. (N-normal, M-mucinous, E-endometrioid, C-clear cell, S-serous). In (**E**), the red color indicates serous_high grade sample, blue color for normal sample. In (**F**), red color indicates advanced serous sample, blue color for early serous sample.

**Figure 3 cancers-16-01781-f003:**
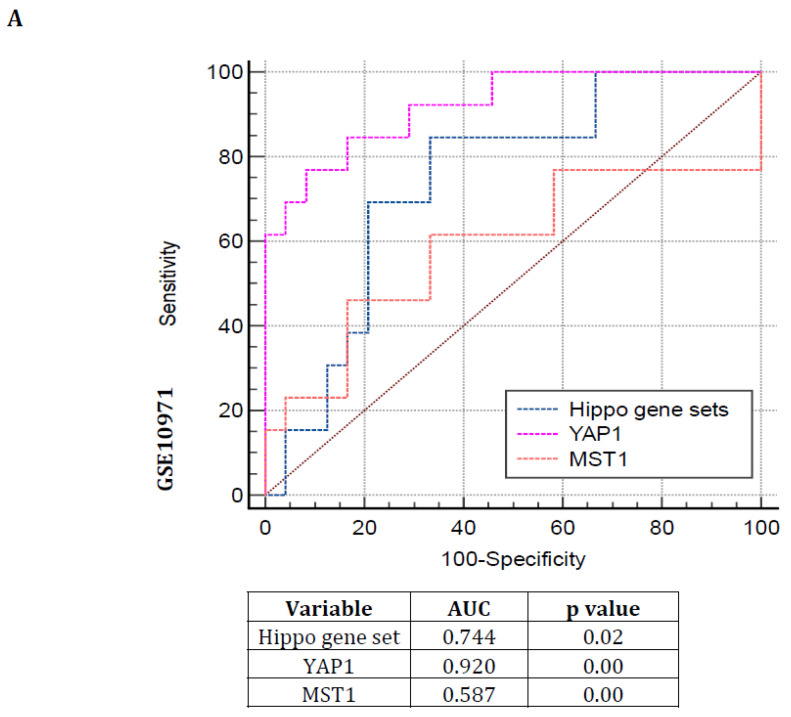
ROC curve-based Hippo signaling pathway dysregulation was assessed in the mRNA expression profiles GSE10971 and GSE14407. (**A**,**B**) ROC curve of Hippo gene set, YAP1, and MST1 genes can predict the serous subtype by greater specificity and sensitivity with significant areas under the curve and *p* values.

**Figure 4 cancers-16-01781-f004:**
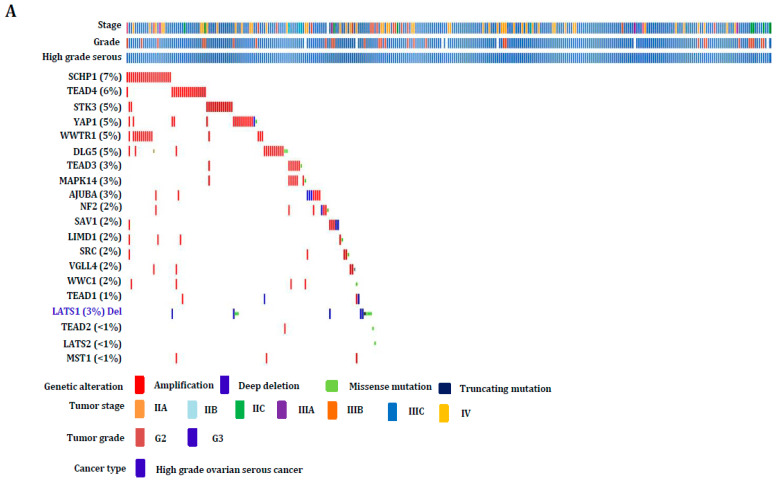
The Hippo gene set was analyzed for its mutation impacts on the profile of high-grade serous ovarian adenocarcinoma. (**A**,**B**) The result of OncoPrint revealed that these genes have been amplified vastly across high-grade serous cancer patients except the LATS1 gene. Moreover, these genes play vital roles, especially in grade three and stage three or four serous-type ovarian tumors.

**Figure 5 cancers-16-01781-f005:**
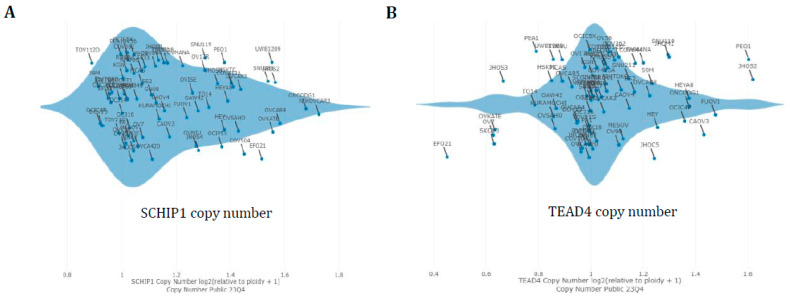
Validation of amplification of Hippo signaling pathway genes in the Dependency Map Portal for ovarian cancer cell lines of copy-number variation analysis. Top amplified genes were studied in this portal. (**A**–**F**) The results of the DepMap plot clearly show that these genes are amplified highly across the ovarian cancer cell lines.

**Figure 6 cancers-16-01781-f006:**
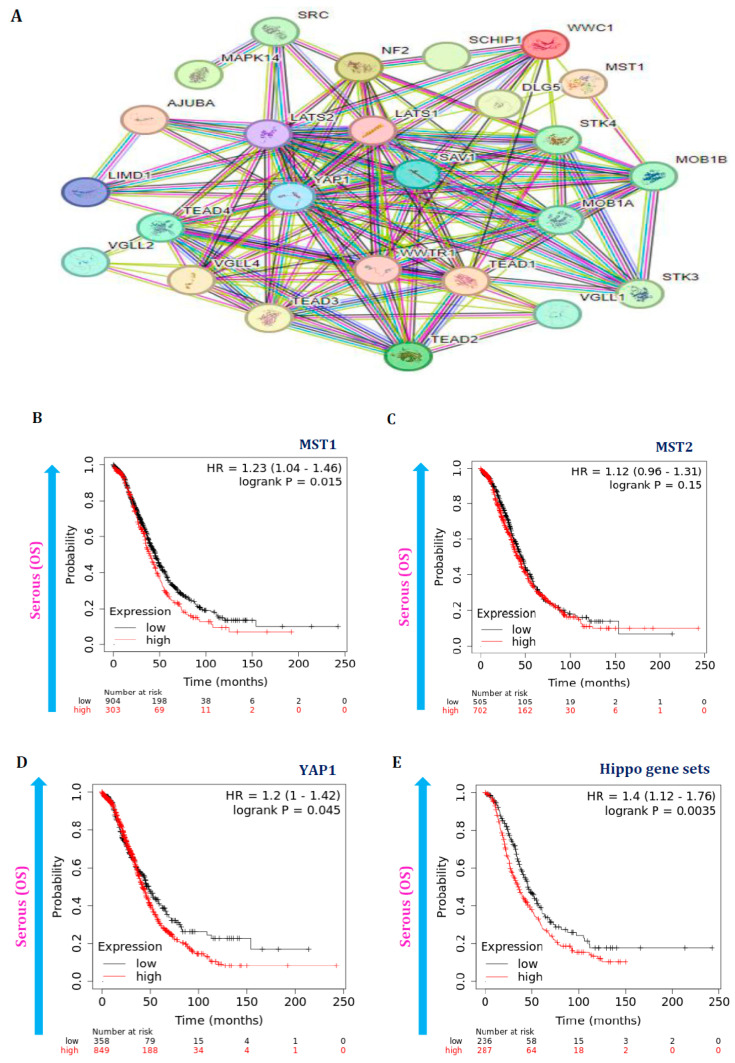
Protein-protein interactions were examined using the STRING database for this gene set. (**A**) The outcome strongly correlates with these proteins and has higher co-expression patterns. Additionally, the impact of these signaling pathway gene expressions on the prognosis of patients with serous carcinoma was investigated. (**B**–**E**) Kaplan-Meier (KM) plots were used to analyze the overall survival (OS) probability in these patients. In patients with serous subtypes, the OS plots of the YAP1 gene and Hippo gene set expressions show a poor survival pattern when highly expressed conditions, while MST1 and MST2 gene expressions do not affect survival. The log-rank test determined the significant *p*-values (*p* < 0.05).

## Data Availability

The original data presented in the study are in a publicly accessible repository, and the data presented in this study are available upon request from the corresponding author for valuable reasons.

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
