# Peer review of "Amplification of Hippo Signaling Pathway Genes Is Governed and Implicated in the Serous Subtype-Specific Ovarian Carcino-Genesis"

_cancers, 2024, doi:10.3390/cancers16091781_

Round 1

Reviewer 1 Report

Comments and Suggestions for Authors

In the article “Amplification of Hippo signaling pathway genes is governed and implicated in the serous subtype-specific ovarian carcino- genesis” authors evaluated, through  integrative functional genomic approaches, the dyregulations of Hippo signaling pathway across the various subtypes comprising ovarian cancer mRNA expression profiles and their cell line profiles demonstrating that the majority of Hippo signalling pathways and their downstream target pathway signatures were highly dysregulated in serous ovarian tumours.

This is a useful and original topic that makes the proposed paper attractive and worthy of consideration. The proposed paper is readily understandable because it is well-constructed, clear and well described with figures exhaustive and appropriate to the subject matter. The study methodology is sound and rigorous and ensures that the research provides valid conclusions. The information collected in this paper could have important implications for a better understanding of the tumor biology and for identifying new molecular mechanisms underlying cancer. The topic falls within the scope of the subject area of the journal.

In my opinion, this work is acceptable for publication after minor revisions which will help the authors to improve the quality of their manuscript:

- I would talk about YAP/TAZ (what they are, how they work, what role they play) in the introduction paragraph, in order to introduce and make the topic immediately clear.

- What is the role of YAP/TAZ in cancer? In which tumours has it already been evaluated? please cite and discuss the following work (https://doi.org/10.1016/j.critrevonc.2021.103246).

- Is there any other evidence on the role of YAP/TAZ in ovarian cancer? If yes, does your data agree with the literature?

Please discuss these data in the discussion section

Reviewer 2 Report

Comments and Suggestions for Authors

An elegant study by Dr. Dong and group elaborating on the role of Hippo signaling in ovarian cancer to develop  the targeted therapies treatment for serous subtype ovarian carcinogenesis. This study has translational implications for determining novel therapeutic regimens. Though a few things must be addressed before it is ready for acceptance, they are as follows:

1. Recently, several studies have shown that Hippo signaling is involved in therapy resistance, including cytotoxic drugs and small-molecule inhibitors (PMID: 37729426 and PMID: 28416665). Authors must add a line mentioning this in the introduction, referring to the relevant articles.

2. It has been known that KRAS mutation or amplification plays a significant role in ovarian cancer (PMID: 36451660 and PMID: 33870211).

Authors must add a comment on their perspective on Hippo's role in ovarian cancer in association with oncogenic KRAS signaling. This aspect will add translational benefit to this study. Authors should add a line in the discussion mentioning this aspect.

3. Several small-molecule inhibitors of Hippo signaling have recently been in clinical trials: Ikena Oncology, Vivace Therapeutics, and Novartis (PMID: 38607003). Authors should mention their point of view on whether those will be beneficial for Hippo-regulated ovarian cancers. 

Round 2

Reviewer 2 Report

Comments and Suggestions for Authors

All concerns addressed and ready for acceptance.